# Serum Osteocalcin in Pediatric Osteogenesis Imperfecta: Impact of Disease Type and Bisphosphonate Therapy

**DOI:** 10.3390/ijms26167953

**Published:** 2025-08-18

**Authors:** Jakub Krzysztof Nowicki, Elżbieta Jakubowska-Pietkiewicz

**Affiliations:** Department of Pediatrics, Neonatal Pathology and Metabolic Bone Diseases, Medical University of Lodz, Sporna 36/50, 91-738 Lodz, Poland; elzbieta.jakubowska-pietkiewicz@umed.lodz.pl

**Keywords:** osteocalcin, osteogenesis imperfecta, bone, children

## Abstract

The aim of this study was to analyze the factors that may influence serum osteocalcin levels in children with osteogenesis imperfecta treated with intravenous sodium pamidronate and to define the role of osteocalcin assessment. The study included 61 patients diagnosed with osteogenesis imperfecta type 1 or 3, aged 2 to 18, hospitalized for intravenous sodium pamidronate administration. A retrospective analysis of medical records was conducted, collecting information on age, sex, body weight, height, the number of long bone fractures throughout life, serum levels of osteocalcin, creatinine, alkaline phosphatase, 25(OH)D_3_, and DXA BMD z-scores for the L1–L4 spine segment. The concentration of osteocalcin is higher in patients with osteogenesis imperfecta than the reference ranges for sex and age. Patients diagnosed with type 3 have significantly lower osteocalcin levels compared to patients with type 1. Also, increasing the age-standardized pamidronate cycle rate significantly reduced osteocalcin concentration. The strongest predictor of osteocalcin concentration among the factors studied is the type of osteogenesis imperfecta. L1–L4 BMD value and fracture frequency were unrelated to osteocalcin concentration. Osteocalcin is an important marker of bone formation that should be measured at the beginning of treatment, as its concentration decreases after successive doses of bisphosphonates.

## 1. Introduction

Osteogenesis imperfecta (OI) is a rare bone dysplasia occurring in 1/15,000–20,000 births [1], characterized by reduced bone mineral density, growth deficiency, and an increased risk of fractures throughout a patient’s life. The clinical manifestation of OI usually includes additional features of connective tissue disorders [2].

### 1.1. Pathogenesis of OI

The most common cause of OI is an autosomal dominant mutation in the *COL1A1* and *COL1A2* genes, which encode the α1 and α2 chains of type I collagen, respectively [3]. However, over the past twenty years, significant progress in genetic diagnostics has led to the identification of around 19 other genes that encode proteins responsible for bone mineralization, proper osteoblast differentiation and function, and collagen modifying, folding, and crosslinking. Mutations in these genes also cause OI [4]. Next-generation sequencing (NGS) and dedicated genetic panels, which target not only mutations in *COL1A1* and *COL1A2* but also recessive mutations in genes like *KBP10*, *PPIB*, *SERPINF1*, *WNT1*, *BMP1*, *CRTAP*, *P3H1*, *SERPINH1*, and *IFITM5*, are becoming increasingly important in the genetic diagnosis of OI [5]. It is important to note that OI can also be caused by mutations linked to the X chromosome [6,7].

### 1.2. OI Classification

In clinical practice, the most commonly used classification of OI is the one proposed by Sillence et al. in 1979, with subsequent modifications [8,9]. This model includes five main types of OI defined by phenotypic characteristics. Current nomenclature guidelines also refer to earlier divisions, but to differentiate from the classical Sillence classification, Arabic numerals are used instead of Roman numerals to designate the OI types [10]. The classification of the patient into a group is made prior to the initiation of treatment.

Patients with type 1, referred to as non-deforming OI with blue sclerae [10], are characterized by an increased risk of bone fractures and reduced bone mineral density without fractures occurring in prenatal life or shortly after birth. Type 2, also known as severe perinatal OI [10], typically results in stillbirth or death in early infancy. However, thanks to invasive respiratory support and intensive care, the adverse prognosis may change [11]. In OI type 2, bone fractures occur prenatally, and the bone deformities are serious [12]. Patients with type 3, or progressively deforming OI [10], experience numerous fractures, which may occur before birth or in early childhood. Bone deformities are present from birth and worsen with age. The last type proposed in the original classification is type 4, or common variable OI with normal sclerae in adult life [10], which represents a phenotype intermediate between types 1 and 3. Patients typically have normally colored sclerae, experience increased fractures, but bone deformities are less frequent compared to type 3 OI [13]. In addition to the aforementioned types, type 5, or osteogenesis imperfecta with calcification in interosseous membranes with or without hyperplastic callus [10], is also widely recognized. This type is characterized by progressive calcification of the interosseous membranes in the forearms and lower legs [14].

### 1.3. Treatment

The treatment of patients with congenital bone fracture requires an individualized approach. It consists of both orthopedic interventions, rehabilitation, psychological care, and pharmacotherapy. Of the drugs used, intravenous bisphosphonates, especially nitrogen-containing, i.e., sodium pamidronate and zoledronic acid, are the standard of treatment. They work by inhibiting farnesyl pyrophosphate synthase (FPPS), which is important in promoting attachment of the osteoclast to the bone, thus reducing bone reabsorption [15] (Figure 1). Single studies have proven the efficacy of therapy among children with OI, also in terms of reducing fracture rates, improving DXA score, and increasing mobility; however, these results were not confirmed in a meta-analysis [16,17,18,19]. Other drugs whose efficacy in the treatment of OI is under investigation are monoclonal antibodies directed against sclerostin (setrusumab and romosozumab) and, to a limited extent, the anti-RANKL antibody—denosumab [20,21,22].

### 1.4. Bone Mineral Density and Bone Formation Markers

One of the fundamental methods for monitoring disease progression and effects of treatment in children with OI is measuring bone mineral density (BMD) using dual-energy X-ray absorptiometry (DXA) [23]. Measurements can be taken for the whole body (total body) or for the vertebrae of the spine (Spine; L1–L4). The results should be interpreted relative to the patient’s sex, age, height, and weight [24]. The presence of orthopedic hardware in the examined bone area may limit the accurate interpretation of DXA results [25].

To assess bone metabolic activity, it is possible to measure blood markers. Commonly tested markers of bone formation in clinical practice are total alkaline phosphatase (ALP) and total osteocalcin (OC), with their concentrations depending on factors such as sex and age [26]. There are reports indicating that OC levels may also depend on serum creatinine concentrations [27,28].

Osteocalcin, otherwise known as bone γ-carboxyglutamic acid protein, is produced by osteoblasts during the bone formation process and is incorporated in a carboxylated form into the bone matrix. As a result of osteoclast activity, bone reabsorption, and pH reduction, osteocalcin is decarboxylated and released into the bloodstream [29] (Figure 1). In addition, it is postulated that uncarboxylated OC may have hormone-like activity and affect carbohydrate metabolism, nervous system development, and male fertility [30].

Serum OC levels are widely used as a bone formation marker, especially among patients with metabolic bone diseases. However, data on factors affecting OC levels are limited, particularly in patients with rare diseases undergoing specialized treatments, which affects the interpretation of the results and their utility in the clinical assessment of patients.

The aim of this study was to analyze the relationship between serum OC levels in patients with OI types 1 and 3 treated with pamidronate sodium in a single-center Polish cohort and other clinically significant parameters, and to define the role of OC assessment of children in the above group in daily clinical practice.

## 2. Results

In our cohort, the z-score of OC concentration, adjusted for sex and age norms, was 2.37, indicating that the OC levels of our patients were higher than the reference ranges for sex and age.

Statistically significant differences were found between patients with OI type 1 and type 3 in the studied variables, including the z-score values of OC concentration in serum (Table 1). In patients with OI type 1, the median OC concentration z-score was 4.27, which was significantly higher than in patients with OI type 3 (0.70; *p* < 0.001) (Figure 2). Similar trends were observed for the z-score values of ALP concentration in serum (OI type 1 vs. OI type 3: −0.92 vs. −1.43; *p* = 0.04, respectively).

Patients with OI type 1 also had higher bone mineral density (BMD) z-scores for the L1–L4 vertebrae (−0.35 vs. −3.14; *p* < 0.001) and a lower fracture rate per age (0.56 vs. 1.91; *p* < 0.001). The anthropometric measurements between the groups showed significant differences in height and weight z-scores (patients with OI type 1 were taller and heavier), while no significant difference in BMI z-score was found. Patients with OI type 3 received higher doses of calcium and 25(OH)D_3_ per kilogram of body weight (*p* < 0.001 in both comparisons). These patients also achieved higher mean serum 25(OH)D_3_ concentrations (t.1 vs. t. 3; 36.49 ng/mL vs. 40.28 ng/mL), but this difference was not statistically significant (*p* = 0.21). The average serum concentration of 25(OH)D_3_ in the entire study cohort remained within optimal values. The difference in the average serum calcium concentration was negligible (2.48 mmol/L vs. 2.49 mmol/L). There were also no significant differences in glucose concentration nor the GFR between the groups (Table 1).

Analyzing the relationships between the z-score of serum OC concentration and other continuous variables, significant statistical correlations were found. The OC z-score was negatively correlated with fracture rate per age (weak correlation) and the number of pamidronate sodium cycles per age (moderate correlation). It was positively correlated with the z-score of serum ALP concentration (weak correlation), BMD L1–L4 z-score, and height and weight z-scores (moderate correlations). We did not find correlations between OC and BMI z-score, glucose, 25(OH)D_3_, calcium concentrations, nor GFR (Table 2, Figure 3).

In single-factor linear regression models, it was demonstrated that disease type, number of pamidronate sodium cycles per age, and number of bone fractures per age were associated with significantly lower osteocalcin concentrations, while ALP concentration, body weight z-score, growth z-score, and BMD L1–L4 z-score were associated with significantly higher osteocalcin concentrations (Table 3).

Single-factor models were also evaluated in subgroups depending on the type of OI, but they did not reveal any statistically significant associations.

Due to the presence of significant correlations within the parameters studied, in order to reduce the dimensions of the analysis, a principal component analysis (PCA) was performed, in which one significant dimension describing the variables studied was observed. Dimension one consisted of the z-score of mass and the z-score of height. The factor describing dimension one (PC1) was included in the base and further measurements.

Considering the above, a multivariate linear regression model was developed to assess the factors affecting OC concentration in patients with OI, where the dependent variable was the z-score of serum OC concentration (Table 4). The created model was statistically significant (*p* < 0.0001), met the assumption of homoscedasticity (p(F) = 0.74), and explained approximately 39.3% of the variability in the OC z-score (adj. R^2^ = 0.393) (Table 4).

The regression analysis showed that the strongest predictor of osteocalcin concentration is the type of OI. Patients diagnosed with type 3 OI have significantly lower osteocalcin levels compared to patients with type 1. Also, increasing the age-standardized pamidronate cycle rate significantly reduced, while ALP concentration increased OC concentrations (Table 4).

At no point in the statistical analysis was a relationship observed between the OC concentration z-score and GFR or serum 25(OH)D_3_ concentration.

## 3. Discussion

Children with OI have higher osteocalcin concentrations than the unaffected population. Similar observations have been reported in adult cohorts with OI [31,32]. OC, as the most abundant non-collagenous protein in bone, is produced by osteoblasts during bone formation [33,34]. In a study by Rauch et al., histomorphometry showed that patients with OI had, on average, a 2.2 to 3.7 times larger osteoblast surface area in bone tissue than the healthy control group, which may explain the higher serum OC levels observed in patients with OI [35].

In the study by Braga V. et al. [32], which included 77 adult patients with OI, 58 of whom were diagnosed with type 1 and 7 with type 3, it was found that patients with type 1 OI had an average OC concentration of 18.7 ng/mL, while those with type 3 had only 14.8 ng/mL. The article did not comment on the statistical significance of the difference in OC concentrations between OI types, but it did demonstrate that the average OC level for all OI patients was statistically significantly higher than that of the control group (17.5 vs. 13.2 ng/mL, *p* < 0.001).

A similar analysis of bone metabolism markers in children with OI was performed by Åström E. et al. in 2010 [36]. The study included 130 patients under 18 years of age with diagnosed OI, who had not previously been treated with bisphosphonates. The median serum OC concentration in patients with OI type 1 was 110 µg/L, while those with OI type 3 had a median of 40.5 µg/L. The study also included patients with OI type 4, who had a median OC concentration of 54.0 µg/L. The difference between OC concentrations in patients with OI type 1 versus types 3 and 4 was statistically significant (*p* < 0.001). [36]

The results of our study, which focused exclusively on patients currently being treated with sodium pamidronate, correspond to the findings reported by Braga V. et al. [32] and Åström E. et al. [36]. Interestingly, Åström’s study [36] also included patients with type 4 OI, whose phenotype severity lies between types 1 and 3, similarly to the median OC levels in this publication. Comparable findings have also been reported in adult populations with OI who had never been treated with bisphosphonates [36].

Histomorphometric studies have demonstrated significantly higher values of bone surface per bone volume in the bone tissue of patients with OI type 3 compared to type 1 (27.7 vs. 19.8 mm^2^/mm^3^), and a higher osteoblast surface per osteoid surface ratio (53% vs. 39%) [35]. These findings suggest greater impairment of osteoblast activity in OI type 3 than in type 1. This may be related to the increased cell surface expression of receptors for transforming growth factor-β (TGF-β) on osteoblasts, which is a known factor leading to defective bone mineralization and severe hypoplasia of long bones, observed in OI patients [37,38,39,40]. The relationship between phenotype severity and TGF-β receptor expression requires further study.

Bisphosphonates, by acting as an antiresorptive, reduce the action of osteoclasts on the bone matrix and therefore reduce the release of uncorboxylated OC into the bloodstream, thereby resulting in lower serum total OC concentrations [14] (Figure 1). We observed a similar treatment effect among the patients included in the present analysis—an increase in the number of pamidronate cycles per age predisposed to lower serum OC concentrations. This observation, which is consistent with studies conducted among adults taking antiresorptive treatment, is an important factor limiting the use of OC as a marker for clinical evaluation among pediatric patients taking long-term bisphosphonates [41,42].

An interesting observation is the correlation we found between BMD L1–L4 z-scores and serum OC z-scores. Our analysis demonstrated a positive correlation between these markers (Table 2). Similar results were obtained in a study by Zhen WB et al. from 2022 [43], which included 225 children with OI and found a positive correlation between OC levels and lumbar spine BMD z-scores (r = 0.258, *p* < 0.001). The study also reported a positive correlation between OC and ALP levels in serum (r = 0.271, *p* < 0.001), consistent with our findings. Zhen WB’s study [43] focused largely on the role of OC in regulating glycolipid metabolism in children with OI and stated that OC levels were significantly associated with the patient’s BMI and fasting blood glucose (FBG) levels, which we did not observe in our analysis. This discrepancy may be due to differences in the inclusion criteria—Zhen WB’s study included patients who had never received antiresorptive therapy such as bisphosphonates, suggesting a milder disease phenotype among the study participants. Additionally, both Zhen WB’s study [43] and ours refer to populations with a specific ethnic background, which may account for additional differences in the findings. The size of the studied cohorts may also play a role.

In the study cited above [43], a negative correlation was also observed between the number of bone fractures and OC levels in serum (r = −0.237; *p* = 0.001). In our analysis, we used the number of fractures standardized for age and likewise observed a negative correlation between this variable and the OC z-score.

The above observations, i.e., the correlation between OC levels and L1–L4 BMD score and fracture rate, did not prove statistically significant in the multivariate linear regression model we developed. In the cited study [43], the regression model was only used to assess the correlation between serum OC levels and glycolipid metabolic parameters, which was not the main focus of our study. Given that both the incidence of long bone fractures and BMD score are closely related to the type of OI, correlations not supported by a statistically significant regression result should not be considered clinically valid. It should also be noted that OC levels rise directly after a fracture, and elevated levels can persist in serum for up to 6 months [44].

Despite the significant findings of this study, several limitations should be considered when interpreting the results.

First, the relatively small sample size (n = 61) limits the generalizability of the results to a broader population of children with OI. However, it provides a useful reference point for the Polish population, given the rarity of the disease and the fact that not all children with OI qualify for bisphosphonate treatment. The study focused solely on patients with OI types 1 and 3, excluding those with other types, which further reduced the study group size. This was due to the small population of OI type 4 and other rarer OI types treated at our center. Additionally, patients whose phenotype did not allow for clear classification into a specific OI type were not included in the study.

Second, the study’s retrospective nature involves limited control over the data collected. Furthermore, due to the retrospective design, we were unable to plan a long-term follow-up of the patients, which prevented us from assessing dynamic changes in OC levels, especially in relation to the administration of subsequent cycles of pamidronate sodium.

Finally, data regarding only BMD L1–L4, and not total body BMD, were collected due to the risk of data distortion caused by rods used to secure the bones of some patients.

These limitations suggest the need for further prospective studies to deepen the analysis on larger groups of patients.

## 4. Methodology

A retrospective review of medical records was conducted for patients hospitalized in the Department of Paediatrics, Neonatal Pathology, and Metabolic Bone Diseases at the University Pediatric Center, Central Clinical Hospital of the Medical University of Lodz, for intravenous pamidronate administration with diagnosed OI between 1 January 2023, and 1 September 2024.

All procedures performed in this study involving human participants were in accordance with the ethical standards of the Ethics Committee of the Medical University of Lodz and the 1964 Helsinki declaration and its later amendments or comparable ethical standards. Ethical approval for all experimental protocols was applied for and granted by the Ethics Committee of the Medical University of Lodz (RNN/173/25/KE).

All patients and their legal guardians gave their written informed consent to participate in the study.

The inclusion criteria for the study were as follows: age between 2 and 18 years, genetically confirmed OI with mutations in the *COL1A1* and/or *COL1A2* genes, type 1 or 3 of OI, previous administration of at least one cycle of pamidronate sodium, and ongoing treatment with the drug.

The exclusion criteria were as follows: comorbidities significantly affecting development or other metabolic bone diseases, a long bone fracture within six months prior to testing, post-surgical or orthopedic interventions, and incomplete documentation.

Based on the above criteria, a group of 61 patients was selected. From the medical records, the following data were obtained: date of birth, sex, body weight, height, type of OI, number of long bone fractures over the patient’s lifetime, the number of pamidronate sodium cycles administered, serum concentrations of OC, creatinine, ALP, calcium, 25(OH)D_3_, and the DXA z-score for the L1–L4 spine segment. All patients received intravenous sodium pamidronate at a dose of 3 mg/kg during the delivery cycle according to the scheme proposed by Glorieux, F.H. et al. [45]. The measurements were taken at least three months after the last dose of pamidronate sodium. All patients received regular supplementation with cholecalciferol and calcium under control of their serum levels.

Height was measured using a stadiometer in a standing position, while in bedridden children, body length was measured lying down with a measuring tape. Both measurements were accurate to 1 cm. Body weight was measured using a SECA 756 (seca, Hamburg, Germany) among children moving independently, and chair weight was measured using a SECA 956 (seca, Hamburg, Germany) among other patients, with an accuracy of 100 g. The body mass index (BMI) was calculated using the standard formula: BMI = body weight [kg]/(height [m])^2^. These measurements were plotted on age- and sex-appropriate percentile charts. For children aged 2–5 years, WHO percentile charts were used [46], and for children over 5 years, percentile charts developed for the Polish population in the OLA and OLAF projects were applied [47,48,49]. The percentiles were then converted to z-scores.

OC concentration was tested using an electrochemiluminescence immunoassay (Roche, Basel, Switzerland), and 25(OH)D_3_ concentration was tested using a chemiluminescent microparticle immunoassay (Abbott, Dublin, Ireland). Serum levels of creatinine and ALP were measured by automated analyzers (Alinity c, Abbott, Dublin, Ireland).

OC and ALP levels were standardized, taking into account the reference ranges appropriate for the sex and age of the patients. The upper and lower reference values were set at +2 and −2 standard deviations, respectively. The reference ranges for the studied parameters were consistent with the guidelines of the Central Clinical Hospital of the Medical University of Lodz, where the tests were performed. Using the Schwartz formula [50], glomerular filtration rate (GFR) was calculated for the patients based on serum creatinine levels and the patient’s height/length measurement.

### Statistical Analysis

The distribution of variables was assessed using histograms and the Shapiro–Wilk test results. Qualitative variables were described using frequency and percentage. Differences between variables were examined using the Chi-square test. Continuous variables with normal distribution were described using the mean and standard deviation, and differences were tested with the t-test. Non-normally distributed variables were described using the median and the first and third quartiles, with differences examined using the Mann–Whitney U test. The relationships between continuous variables were analyzed using the Spearman correlation coefficient. Based on these models, a multivariate linear regression model was developed.

The statistical analysis was performed using STATISTICA 13 with Kit PLUS v. 5.0.85. A *p*-value < 0.05 was considered statistically significant.

## 5. Conclusions

OI type is the most frequent determinant of OC concentrations among the factors studied. More effective bone formation observed in type 1 OI results in higher concentrations of osteocalcin than in type 3. The use of pamidronate sodium decreases patients’ serum OC levels in the long term. Neither L1–L4 BMD value nor fracture frequency had a statistically significant effect on OC concentration.

OC is an important marker of bone metabolism that should be determined in patients with OI, especially at the beginning of treatment, but the need for continuous monitoring of this factor appears to be limited.

## Figures and Tables

**Figure 1 ijms-26-07953-f001:**
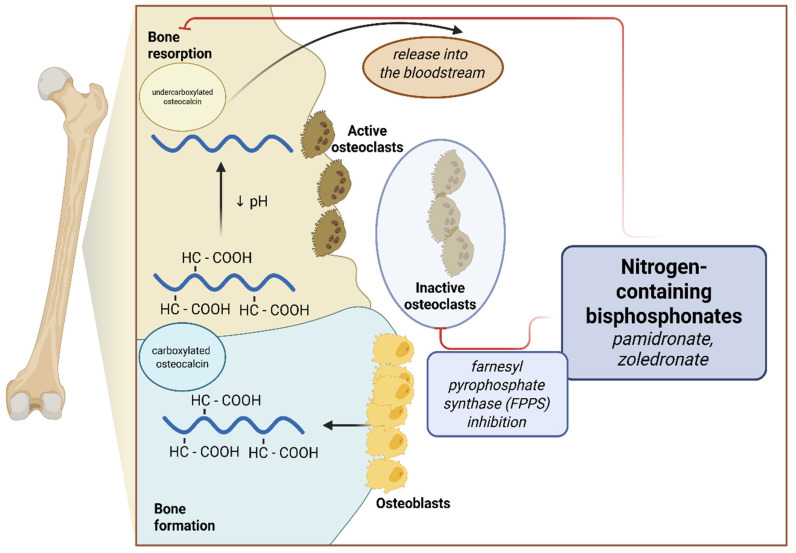
Effects of nitrogen-containing bisphosphonates on bone metabolism and osteocalcin secretion.

**Figure 2 ijms-26-07953-f002:**
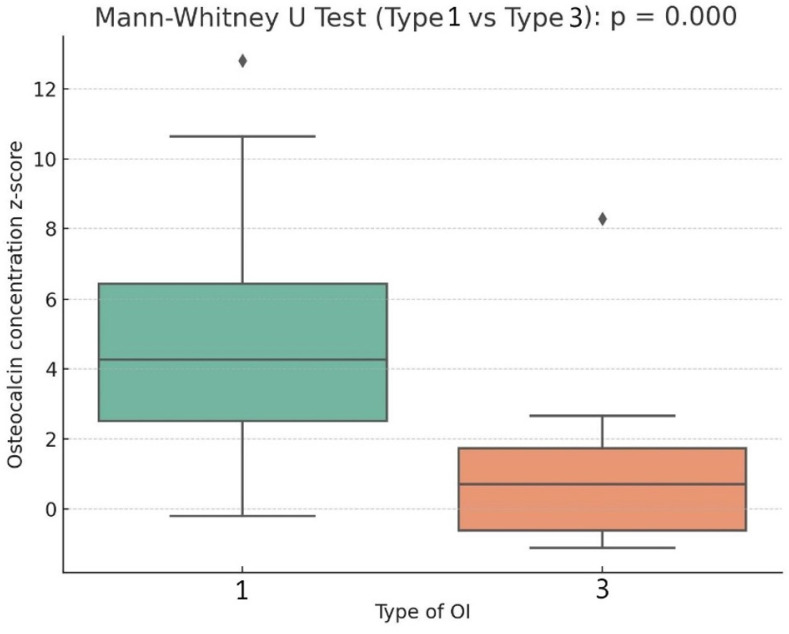
Box plot describing the relation of osteocalcin concentration z-score to type of OI.

**Figure 3 ijms-26-07953-f003:**
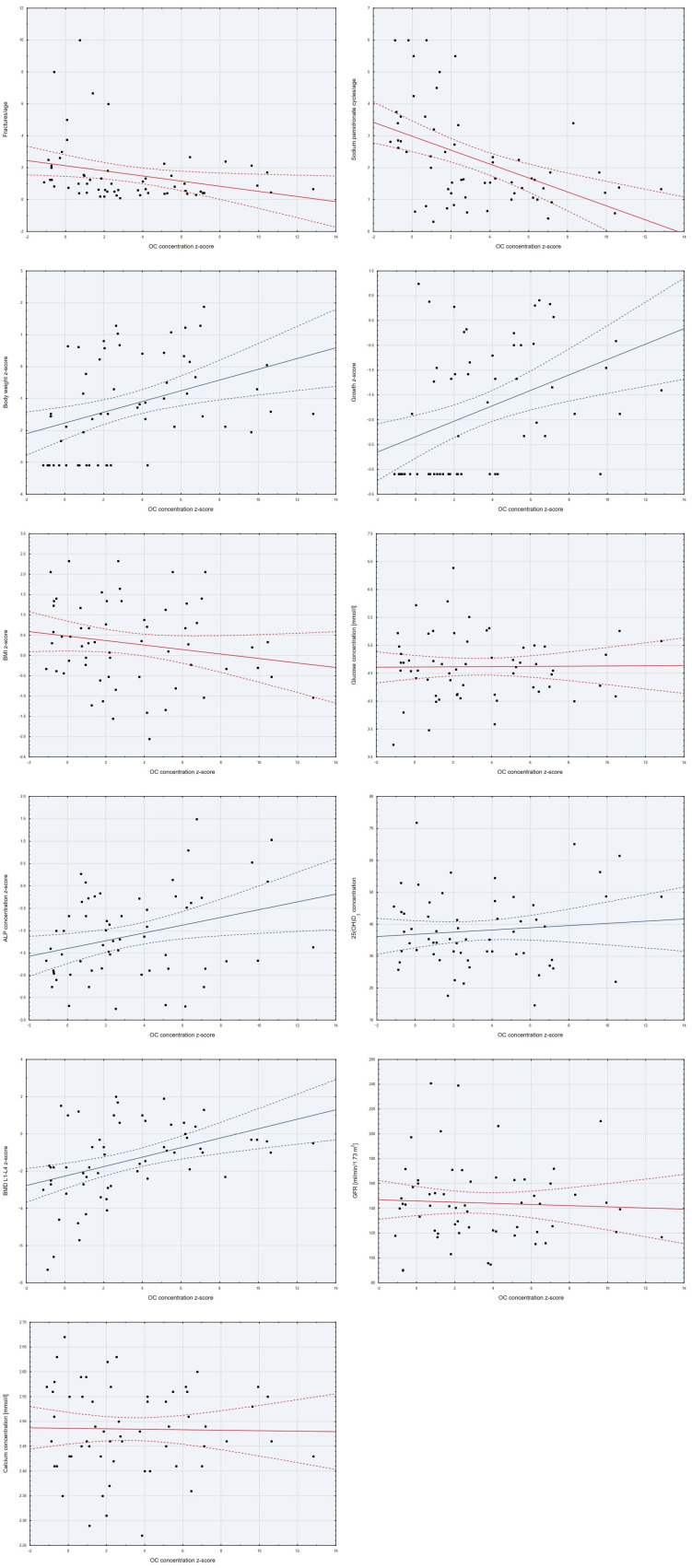
Scatter plots of the variables studied with the z-score of OC concentration.

**Table 1 ijms-26-07953-t001:** Characteristics of the variables.

Variable	Total *n* = 61	Type 1 *n* = 37	Type 3 *n* = 24	*p* t. 1 vs. t. 3
Age [years]	8.0 (5.0; 12.0)	10.0 (6.0; 13.0)	5 (3.5; 8.0)	*p*^UMW^ = 0.002
Sex [F/M]	30/31	17/20	13/11	*p*^Chi2^ = 0.53
Number of fractures of long bones/age	1.00 (0.50; 2.00)	0.56 (0.40; 1.00)	1.91 (1.04; 3.44)	*p*^UMW^ < 0.001
Sodium pamidronate cycles/age	1.67 (1.20; 3.13)	1.35 (0.92; 1.64)	3.37 (2.80; 4.31)	*p*^UMW^ < 0.001
Body weight z-score *	−1.48 (−3.09; 0.19)	−0.33 (−1.28; 0.61)	−3.09 (−3.09; −1.64)	*p*^UMW^ < 0.001
Growth z-score *	−1.97 (−3.09; −0.77)	−1.08 (−1.88; −0.25)	−3.09 (−3.09; −3.09)	*p*^UMW^ < 0.001
BMI z-score *	0.3 (−0.41; 1.15)	0.33 (−0.52; 1.13)	0.23 (−0.33; 1.17)	*p*^UMW^ = 0.8
Glucose concentration [mmol/L]	4.62 ± 0.57	4.71 ± 0.53	4.46 ± 0.63	*p*^t-student^ = 0.1
OC concentration z-score ˟	2.37 (0.74; 5.48)	4.27 (2.52; 6.44)	0.70 (−0.65; 1.77)	*p*^UMW^ < 0.001
ALP concentration z-score ˟	−1.12 ± 0.96	−0.92 ± 1.06	−1.43 ± 0.67	*p*^t-student^ = 0.04
25(OH)D_3_ concentration [ng/mL]	37.98 ± 11.56	36.49 ± 10.77	40.28 ± 12.58	*p*^t-student^ = 0.21
Calcium concentration [mmol/L]	2.48 ± 0.09	2.48 ± 0,09	2.49 ± 0.08	*p*^t-student^ = 0.87
GFR [mL/min/1.73 m^2^]	142 (121; 160)	133 (121; 150)	151 (135; 171)	*p*^UMW^ = 0.05
BMD L1–L4 z-score *	−1.45 ± 2.04	−0.35 ± 1.46	−3.14 ± 1.62	*p*^t-student^ < 0.001
Calcium supplementation [mg/kg/day]	9.1 (6.3; 14.4)	7.1 (5.1; 9.1)	15.3 (11.2; 17.4)	*p*^UMW^ < 0.001
25(OH)D_3_ supplementation [IU/kg/day]	52.6 (35.7; 83.6)	42.4 (30.3; 65.6)	75.9 (55.6; 125.0)	*p*^UMW^ < 0.001

*—adapted by sex and age; ˟—adapted by reference ranges considering sex and age; UMW—U Mann–Whitney test; BMI—body mass index, OC—osteocalcin, ALP—alkaline phosphatase, BMD—bone mineral density, GFR—glomerular filtration rate.

**Table 2 ijms-26-07953-t002:** Spearman correlation values of the osteocalcin concentration z-score variable with other continuous variables.

Variable	Spearman Correlation	*p*-Value
Fractures/age	−0.354	=0.005
Sodium pamidronate/age	−0.525	<0.0001
Body weight z-score	0.470	<0.001
Growth z-score	0.496	<0.001
BMI z-score	−0.163	=0.300
Glucose concentration [mmol/L]	−0.021	=0.873
ALP concentration z-score	0.269	=0.036
25(OH)D_3_ concentration [ng/mL]	0.012	=0.927
Calcium concentration [mmol/L]	−0.06	=0.680
BMD L1–L4 z-score	0.478	<0.001
GFR [mL/min/1.73 m^2^]	−0.044	=0.737

BMI—body mass index; BMD—bone mineral density; ALP—alkaline phosphatase; GFR—glomerular filtration rate.

**Table 3 ijms-26-07953-t003:** Single-factor linear regression models assessing the impact of variables on osteocalcin concentration.

Variable	Estimation	Standard Error	*p*-Value	95% Cl Lower	95% Cl Upper
Type of OI (ref.—OI type 1)	−0.013	0.002	0	−0.017	−0.008
Age [years]	0.018	0.012	0.157	−0.007	0.042
Sex (ref.—female)	−0.001	0.001	0.389	−0.004	0.002
Fractures/age	−0.012	0.005	0.022	−0.023	−0.002
Sodium pamidronate cycles/age	−0.017	0.004	0	−0.024	−0.009
Body weight z-score	0.378	0.273	0.003	0.302	1.393
Growth z-score	0.407	0.314	0.001	0.438	1.695
BMI z-score	−0.18	0.424	0.168	−1.44	0.257
Glucose concentration [mmol/L]	0	0.002	0.99	−0.003	0.003
ALP concentration z-score	0.005	0.003	0.041	0.014	0.685
25(OH)D_3_ concentration	0.01	0.033	0.773	−0.057	0.076
BMD L1–L4 z-score	0.017	0.005	0.002	0.006	0.028
GFR [mL/min/1.73 m^2^]	0.007	0.092	0.939	−0.177	0.191
Calcium concentration [mmol/L]	−0.019	0.131	0.887	−10.811	9.372

Cl—confidence interval; OI—osteogenesis imperfecta; ALP—phosphatase alkaline; BMD—bone mineral density.

**Table 4 ijms-26-07953-t004:** Linear regression model for the explanatory variable: osteocalcin concentration z-score. Adjusted R^2^ = 0.393; *p* < 0.0001.

Variable	B	Standard Error	*p*-Value	95% Cl Lower	95% Cl Upper
Type of OI (OI type 1—ref.)	−34.854	16.213	0.036	−67.360	−2.349
Fractures/age	3.640	3.628	0.320	−3.634	10.914
Sodium pamidronate cycles/age	−12.893	5.692	0.028	−24.305	−1.481
ALP concentration z-score	0.194	0.079	0.017	0.036	0.352
BMD L1–L4 z-score	4.323	3.498	0.222	−2.690	11.336
PC1	−0.341	0.178	0.061	−0.698	0.017

Cl—confidence interval; OI—osteogenesis imperfecta; ALP—phosphatase alkaline; BMD—bone mineral density; PC1—z-score of mass and z-score of height.

## Data Availability

The data presented in this study are available on request from the corresponding author due to (specify the reason for the restriction).

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
