# Peer review of "Serum Osteocalcin in Pediatric Osteogenesis Imperfecta: Impact of Disease Type and Bisphosphonate Therapy"

_ijms, 2025, doi:10.3390/ijms26167953_

Round 1

Reviewer 1 Report

Comments and Suggestions for Authors

The manuscript presents important insights into osteocalcin’s role in clinical practice in children with osteogenesis imperfecta treated with bisphosphonates. However, I would like to suggest the following suggestions and comments to further strengthen the manuscript.

  1. The authors should revise the manuscript title to improve clarity and conciseness. A focused title better reflects the core findings and enhances the study impact and readability.

  1. Authors should include a table presenting all variables along with their respective mean and standard deviation values for the total sample and both groups (Type 1 and Type 3).

  1. Page2, line 59 to 61

The sentences lack clarity and require revision to enhance readability and flow.

  1. A detailed explanation of the results presented in Table 1 is required in the result section.

  1. Since calcium plays a crucial role in bone metabolism, including an analysis of serum calcium levels is essential. Assessing calcium levels provides insights into the systemic metabolic environment influencing bone health and contextualizes findings related to osteocalcin or other bone biomarkers. Alterations in calcium levels may reflect relevant physiological or pathological processes. Including this data would enhance scientific rigor and provide a comprehensive understanding.

  1. It is essential to provide information regarding the dietary status of the subjects, as nutritional intake, particularly of calcium, vitamin D, and other bone-related nutrients, can substantially impact bone metabolism and biomarker levels.

  1. Authors should include both correlation and linear regression analyses between serum calcium and osteocalcin levels to clarify their relationship. This will strengthen the overall impact of the study.
  2. It is also recommended to perform linear regression analyses separately for each group in Table 3 using all available variables. This approach identifies group-specific patterns, interactions, or predictive relationships that may be obscured when combining data. It clarifies whether variable relationships differ by group or not. These analyses add depth and rigor to the statistical evaluation of the data.

Author Response

Thank you very much for your inspiring feedback, which will help us improve the quality of our work. Below are references to the comments.
Best regards,

Authors.

The manuscript presents important insights into osteocalcin’s role in clinical practice in children with osteogenesis imperfecta treated with bisphosphonates. However, I would like to suggest the following suggestions and comments to further strengthen the manuscript.

  • The authors should revise the manuscript title to improve clarity and conciseness. A focused title better reflects the core findings and enhances the study impact and readability.

 We have changed title: “Serum Osteocalcin in Pediatric Osteogenesis Imperfecta: Impact of Disease Type and Bisphosphonate Therapy”

  • Authors should include a table presenting all variables along with their respective mean and standard deviation values for the total sample and both groups (Type 1 and Type 3).

In the statistical analysis continuous variables with normal distribution were described using the mean and standard deviation, and differences were tested with the t-test. Non-normally distributed variables were described using the median and the first and third quartiles, with differences examined using the Mann-Whitney U test. We cannot use respective mean and standard deviation values for variables with not-normal distribution.

  • Page2, line 59 to 61. The sentences lack clarity and require revision to enhance readability and flow.

We have change this lines as below:

Type 2, also known as severe perinatal OI (11), is typically resulting in stillbirth or death in early infancy. However, thanks to invasive respiratory support and intensive care, the adverse prognosis may change. (12) In OI type 2 bone fractures occur prena-tally, and the bone deformities are serious (13).

  • A detailed explanation of the results presented in Table 1 is required in the result section.

In our opinion all results presented in Table 1 are already explained and described in result section. We did not want to mention all p values in text, especially differences in anthropometric measurements, because this would reduce the clarity of the text. Below we past the paragraphs from the text, that describe values presented in Table 1:

“In our cohort, the z-score of OC concentration, adjusted for sex and age norms, was 2.37, indicating that the OC levels of our patients were higher than the population references.

Statistically significant differences were found between patients with OI type 1 and type 3 in the studied variables, including the z-score values of OC concentration in se-rum (Table 1). In patients with OI type 1, the median OC concentration z-score was 4.27, which was significantly higher than in patients with OI type 3 (0.70; p<0.001) (Chart 1). Similar trends were observed for the z-score values of alkaline phosphatase concentration in serum (respectively, OI type 1 vs. OI type 3: -0.92 vs. -1.43; p=0.04).

Patients with OI type 1 also had higher bone mineral density (BMD) z-scores for the L1-L4 vertebrae (-0.35 vs. -3.14; p<0.001), as well as a lower fracture rate per age (0.56 vs. 1.91; p<0.001). The anthropometric measurements between the groups showed significant differences in height and weight percentiles (patients with OI type 1 were taller and heavier), while no significant difference in BMI percentiles was found. Pa-tients with OI type 3 received higher doses of calcium and 25(OH)D3 per kilogram of body weight (p<0.001 in both comparisons). These patients also achieved higher mean serum 25(OH)D3 concentrations (t.1 vs. t. 3; 36.49 ng/ml vs. 40.28 ng/ml), but this difference was not statistically significant (p=0.21). The average serum concentration of 25(OH)D3 in the entire study cohort remained within optimal values. The difference in the average serum calcium concentration was negligible (2.48 mg/dl vs. 2.49 mg/dl). There were also no significant difference in GFR between the groups (Table 1).”

  • Since calcium plays a crucial role in bone metabolism, including an analysis of serum calcium levels is essential. Assessing calcium levels provides insights into the systemic metabolic environment influencing bone health and contextualizes findings related to osteocalcin or other bone biomarkers. Alterations in calcium levels may reflect relevant physiological or pathological processes. Including this data would enhance scientific rigor and provide a comprehensive understanding.

We have added to Table 1 the total calcium concentrations in blood serum tested before the administration of sodium pamidronate in a given cycle, and we have also added a sentences in the results section referring to the above. We also make futher analysis of regression models including calcium concentrations.

  • It is essential to provide information regarding the dietary status of the subjects, as nutritional intake, particularly of calcium, vitamin D, and other bone-related nutrients, can substantially impact bone metabolism and biomarker levels.

We obtained additional information on calcium and vitamin D3 supplementation from patients' medical records and included it in Table 1. This is the maximum amount of information we can obtain on calcium intake due to the nature of the study (retrospective review of medical records). In addition, I can add that all our patients with OI are recommended a high-calcium diet.

  • Authors should include both correlation and linear regression analyses between serum calcium and osteocalcin levels to clarify their relationship. This will strengthen the overall impact of the study.

We have calculated both the correlation and the linear regression describing the relationship between osteocalcin concentration and serum calcium concentration.

  • It is also recommended to perform linear regression analyses separately for each group in Table 3 using all available variables. This approach identifies group-specific patterns, interactions, or predictive relationships that may be obscured when combining data. It clarifies whether variable relationships differ by group or not. These analyses add depth and rigor to the statistical evaluation of the data.

We have added new table with single-factor linear regression models assessing the impact of variables on osteocalcin concentration calculated for the whole cohort. We have also evaluated single-factor models in subgroups depending on the type of OI, but they did not reveal any statistically significant associations, which we have mentioned in the text. In our opinion the absence of significant factors influencing osteocalcin concentration within subgroups is an additional argument suggesting that the type of disease is the main determinant of its concentration.

Reviewer 2 Report

Comments and Suggestions for Authors

In the present manuscript, Jakub Krzysztof Nowicki and Elżbieta Jakubowska-Pietkiewicz present a retrospective study assessing serum osteocalcin levels in 61 pediatric patients aged 2-18 years with osteogenesis imperfecta (OI) types 1 and 3 treated with intravenous sodium pamidronate. The authors demonstrate that osteocalcin levels are elevated in children with OI compared to the general population, but significantly lower in patients with type 3 than in those with type 1. They also report that increasing the number of pamidronate treatment cycles significantly reduced osteocalcin concentrations. Among all variables studied, the OI type was the strongest predictor of osteocalcin levels. Notably, fracture frequency and lumbar spine BMD z-scores showed no correlation with osteocalcin concentration. Overall, the authors suggest that osteocalcin is a useful marker of bone formation, particularly when measured early in bisphosphonate therapy.

Major Comments

  1. This is an important retrospective study that contributes to understanding the role of osteocalcin in children with OI. However, similar studies have previously been conducted, and the author must emphasize more clearly the novelty of this work.
  2. The study includes only 61 participants, which limits its statistical power, especially for comparisons between OI types. Drawing strong conclusions from such a small cohort may be premature.
  3. The study does not include long-term follow-up data on changes in osteocalcin levels, making it difficult to assess the progression or clinical implications of the findings.
  4. While comparisons to a reference population are mentioned, a healthy age-matched control group was not included. This limits the strength and reliability of conclusions regarding relative osteocalcin levels.
  5. In the authors’ previous publication, “Impact of Somatic Development and Course of Osteogenesis Imperfecta on FGF23 Levels in Children,” FGF23 levels were negatively correlated with age and BMI. Since FGF23 is secreted by differentiated osteoblasts, it would be valuable to know whether any correlation was observed between FGF23 and osteocalcin levels in this study.

Author Response

Thank you very much for your inspiring feedback, which will help us improve the quality of our work. Below are references to the comments.

Best regards,

Authors.

In the present manuscript, Jakub Krzysztof Nowicki and Elżbieta Jakubowska-Pietkiewicz present a retrospective study assessing serum osteocalcin levels in 61 pediatric patients aged 2-18 years with osteogenesis imperfecta (OI) types 1 and 3 treated with intravenous sodium pamidronate. The authors demonstrate that osteocalcin levels are elevated in children with OI compared to the general population, but significantly lower in patients with type 3 than in those with type 1. They also report that increasing the number of pamidronate treatment cycles significantly reduced osteocalcin concentrations. Among all variables studied, the OI type was the strongest predictor of osteocalcin levels. Notably, fracture frequency and lumbar spine BMD z-scores showed no correlation with osteocalcin concentration. Overall, the authors suggest that osteocalcin is a useful marker of bone formation, particularly when measured early in bisphosphonate therapy.

Major Comments

1) This is an important retrospective study that contributes to understanding the role of osteocalcin in children with OI. However, similar studies have previously been conducted, and the author must emphasize more clearly the novelty of this work.

The novelty presented in this study is the unique selection of the patient cohort. All patients included in the study are exclusively patients of developmental age, which significantly changes the dynamics of bone metabolism. In addition, these are patients treated exclusively with sodium pamidronate, i.e. one type of bisphosphonate, according to a single proposed regimen. This selection makes the cohort homogeneous in terms of both disease and treatment, allowing for a better understanding of the mechanisms determining osteocalcin concentration among these patients. Furthermore, this analysis focuses on comparing osteocalcin concentrations depending on the type of OI, which is its main objective, while other cited studies did not focus on this aspect of osteocalcin concentration assessment.

2) The study includes only 61 participants, which limits its statistical power, especially for comparisons between OI types. Drawing strong conclusions from such a small cohort may be premature.

We are aware of the limitation posed by the number of patients, which we have also mentioned in the section on the limitations of our study. However, OI is a rare disease, with an estimated 300-400 patients diagnosed in Poland, although the exact number is unknown. It should be noted that our cohort included only individuals of developmental age who were taking sodium pamidronate. Considering the above, we find that our cohort is representative of the Polish population of patients of developmental age with OI treated with sodium pamidronate, to which our conclusions refer.

 3) The study does not include long-term follow-up data on changes in osteocalcin levels, making it difficult to assess the progression or clinical implications of the findings.

This is a very important suggestion and inspiration for further work allowing for a better understanding of the mechanisms determining osteocalcin concentration in our patients. We plan to perform the analysis mentioned by the Reviewer in the future, although at this point we do not have sufficient data to do so. However, we believe that this work provides some insight into the clinical implications, especially in the context of the relationship between the number of bisphosphonate cycles patients received and osteocalcin concentration. We also consider the difference between osteocalcin concentrations depending on the type of OI to be significant, as it draws attention to the relationship between the patient's phenotype and the severity of the disease, and the OC concentration itself.

 4) While comparisons to a reference population are mentioned, a healthy age-matched control group was not included. This limits the strength and reliability of conclusions regarding relative osteocalcin levels.

In the text we have used an unfortunate phrase that could indeed suggest a reference to a control group. In this study, also due to its nature as a retrospective review of documentation, we did not refer to a control group, but only to the ranges of norms, which was clarified in the text of the article.

 5) In the authors’ previous publication, “Impact of Somatic Development and Course of Osteogenesis Imperfecta on FGF23 Levels in Children,” FGF23 levels were negatively correlated with age and BMI. Since FGF23 is secreted by differentiated osteoblasts, it would be valuable to know whether any correlation was observed between FGF23 and osteocalcin levels in this study.

We are more than pleased that the previous article, authored by Agnieszka Byrwa-Sztaba and Elżbieta Jakubowska-Pietkiewicz (doi: 10.3390/ijms26136007), has attracted the interest of the Reviewer. I would like to inform you that the relationship between FGF23 and osteocalcin was investigated by the authors of the aforementioned article and proved to be insignificant, in other words, no relationship was found between them. The authors decided not to include this observation in the article. This conclusion also went beyond the scope of this text, partly because FGF23 was not studied in this cohort. However, I have suggested to the authors of the previous article that such observations are of interest to the Reviewer.